# Scalable-DSC: A Structural Template Prompt Approach to Scalable Dialogue State Correction

**Haoxiang Su**[1†]**, Hongyan Xie**[3†]**, Hao Huang**[1,4‡]**, Shuangyong Song**[2]**,**
**Ruiyu Fang**[2]**, Xiaomeng Huang**[2] **and Sijie Feng**[4]

[1]School of Computer Science and Technology, Xinjiang University, Urumqi, China
[2]China Telecom Corporation Ltd. AI Technology Company    [3]JD AI Research, Beijing, China
[4]Xinjiang Provincial Key Laboratory of Multi-lingual Information Technology, Urumqi, China
{haoxisu98, hwanghao, verdantpauline}@gmail.com  xiehongyan1@jd.com
{songshy, fangry, huangxm26}@chinatelecom.cn

## Abstract

Dialogue state error correction has recently been proposed to correct wrong slot values in predicted dialogue states, thereby mitigating the error propagation problem for dialogue state tracking (DST). These approaches, though effective, are heavily intertwined with specific DST models, limiting their applicability to other DST models. To solve this problem, we propose Scalable Dialogue State Correction (Scalable-DSC), which can correct wrong slot values in the dialogue state predicted by any DST model. Specifically, we propose a Structural Template Prompt (STP) that converts predicted dialogue state from any DST models into a standardized natural language sequence as a part of the historical context, associates them with dialogue history information, and generates a corrected dialogue state sequence based on predefined template options. We further enhance Scalable-DSC by introducing two training strategies. The first employs a predictive state simulator to simulate the predicted dialogue states as the training data to enhance the generalization ability of the model. The second involves using the dialogue state predicted by DST as the training data, aiming at mitigating the inconsistent error type distribution between the training and inference. Experiments confirm that our model achieves state-of-the-art results on MultiWOZ 2.0-2.4[△].

## 1 Introduction

Task-oriented dialogue systems are becoming increasingly important in facilitating the daily life of human beings. Dialogue State Tracking module plays an important role in dialogue systems (Wang and Lemon, 2013a), which aims to accurately track the user's goals based on the dialogue history and

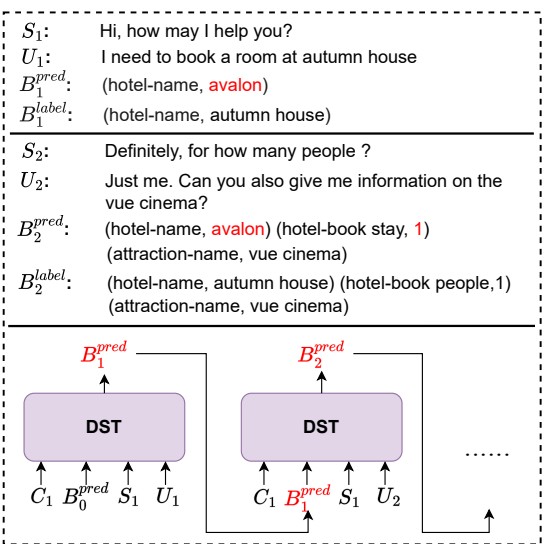

| $S_1$: | Hi, how may I help you? |
| --- | --- |
| $U_1$: | I need to book a room at autumn house |
| $B_1^{pred}$: | (hotel-name, avalon) |
| $B_1^{label}$: | (hotel-name, autumn house) |
| $S_2$: | Definitely, for how many people ? |
| $U_2$: | Just me. Can you also give me information on the vue cinema? |
| $B_2^{pred}$: | (hotel-name, avalon) (hotel-book stay, 1) (attraction-name, vue cinema) |
| $B_2^{label}$: | (hotel-name, autumn house) (hotel-book people,1) (attraction-name, vue cinema) |

Figure 1: An example of error propagation in DST task. "$S_1$" and "$U_1$" refer to the system utterance of the 1-th turn and the user utterance of the 1-th turn, respectively. "$B_1^{pred}$" and "$B_1^{label}$" represent the predicted results of the dialogue state for the first turn model, and the ground truth of the first turn dialogue state. "$C_1$" refers to the first turn of dialogue history. The other symbols can be inferred in sequence. The red characters indicate errors.

represent the dialogue states as a set of (slot,value) pairs.

Currently, some DST approaches (Ye et al., 2021b; Tian et al., 2021; Zhou et al., 2021; Xu et al., 2023) attempt to track the user's goal from the dialogue state of the previous turn, which as a compact representation of the dialogue history information (Kim et al., 2020), and the dialogue utterance for the current turn. These approaches have improved the efficiency of DST tasks. However, the errors generated in the current turn of the model are likely to be carried over to the next turn. Figure 1 illustrates an example of error propagation: the predicted dialogue state $B_1^{pred}$ in the first turn, which contains the wrong slot values, influences the prediction $B_2^{pred}$ of the DST model in the second turn. Thus, recent studies (Tian et al., 2021;

---

[†] Equal Contribution.
[‡] Corresponding author.
[△]Our code is available at https://github.com/xjuspeech/Scalable-DSC

Xie et al., 2022) focus on correcting wrong slot values in predicted dialogue states to mitigate the error propagation. The correction modules of these approaches are deeply intertwined with specific DST models, limiting their applicability to other DST models. Moreover, to mitigate historical context mismatch between training and inference, by utilizing certain strategies, these approaches generate predicted dialogue states which may contain wrong slot values to train the model. However, the inconsistent error distribution between the predicted dialogue states generated by certain strategies during training and the dialogue states predicted by DST during inference is ignored, which limits the correction capability of the model.

To address the above problem, we propose Scalable-DSC. This is a standalone dialogue state correction model whose responsibility is to correct the wrong slot values in the dialogue state predicted by the DST model. Firstly, to enhance the scalability of Scalable-DSC, we propose the Structural Templates Prompt (STP) approach, which consists of four components in its input schema: (1) *Dialogue History* provides Scalable-DSC with real dialogue information, serving as crucial evidence to correct the dialogue state; (2) By using heuristic scripts, the predicted dialogue state of any DST model is transformed into a natural language sequence (*State Sequence*), which is considered as part of the model's historical context, and then a dialogue state sequence is generated with corrected slot values. (3) *Slot Options* enable the model to associate predefined template content with relevant slot descriptions, making the model aware of which slot information needs to be selected; and (4) *Template Options* prompts the model to select relevant template content for performing controlled conditional generation. This generation approach forces Scalable-DSC to firstly generate a sequence of erroneous states and then continue generating a sequence of corrected states. Secondly, we employ two training strategies to optimize the Scalable-DSC model in a staged manner. Strategy 1: During training, we use a predictive state simulator (Xie et al., 2022) to dynamically generate predicted dialogue states, which contain different wrong slot values in each epoch to enhance the model's generalizability. Strategy 2: We use the DST model to predict the dialogue state as the training data for the Scalable-DSC model. In addition, to provide predictive state data during the Scalable-DSC

model training, we propose a DST model based on the STP approach (STP-DST), as shown in Figure 2.

We conducted extensive experiments on MultiWOZ 2.0-2.4(Budzianowski et al., 2018; Eric et al., 2020; Zang et al., 2020; Han et al., 2021; Ye et al., 2021a) , and the results indicate that Scalable-DSC significantly improves the performance of STP-DST by correcting dialogue state errors and achieves a new state-of-the-art. The contributions of the paper are as follows:

(1) We propose Scalable-DSC, a standalone dialogue state correction model, that can universally correct the predicted results of different DST models, adaptively infer errors in the dialogue state sequence and then generate a complete and correct sequence.

(2) We introduce a Structural Template Prompting (**STP**) approach. It controls Scalable-DSC through structured prompt texts to determine what needs to be corrected, what needs to be associated, and what needs to be generated through the template prompting mechanism.

(3) We use predicted dialogue states generated by a predictive state simulator and predicted dialogue states from the DST model to train Scalable-DSC in stages. This approach not only enhances the model's generalization ability but also alleviates the issue of inconsistent distributions between training and inference error types.

## 2 Methodology

Figure 2 illustrates the overall framework of correcting erroneous slot values in dialogue states using Scalable-DSC. In the preprocessing section, Strategy 1 employs a predictive state simulator to simulate predicted dialogue states in the training set. Strategy 2 utilizes a DST model to generate predicted dialogue states in the training set. In the training section, we utilize the predicted dialogue states $B_t^{pred\_seq}$ generated from the preprocessing section as a part of the input schema for the STP method in Scalable-DSC, resulting in the generation of the corrected target sequence $B_t^{error\_seq} \oplus B_t^{correct\_seq}$. In the inference section, Scalable-DSC corrects the prediction results of the DST model on the test set.

### 2.1 The Input Schema of STP Approach

Figure 2 illustrates the input schema of the STP approach in Scalable-DSC. The schema includes

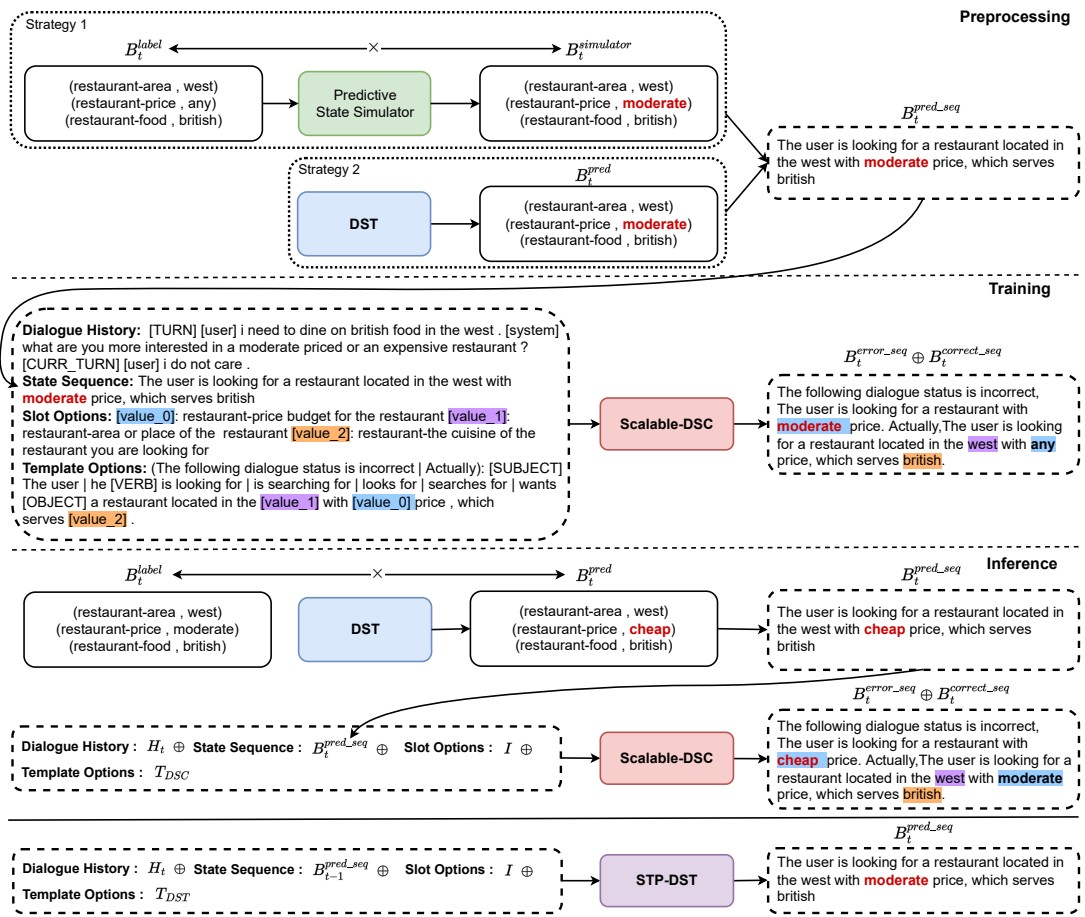

Figure 2: The overview framework of the Scalable-DSC model for correcting dialogue state errors. The red font "moderate" indicates an incorrect slot value in the dialogue state. There is a correlation between the text segments with the same background color.

*Dialogue History*, *State Sequence*, *Slot Options*, and *Template Options*. Each element is described as follows:

**Dialogue History**: Due to the fact that the dialogue history provides real information for generating dialogue states. We use *Dialogue History* aids for Scalable-DSC to correct dialogue states that contain erroneous slot values. For STP-DST, it relies on the dialogue history and the previous dialogue state to predict the current dialogue state. Dialogue History for the $t$-th turn is denoted as $H_t = D_1 \oplus D_2 \oplus \cdots \oplus D_{t-1} \oplus [CURR\_TURN] \oplus [user] \oplus U_t$. $D_t = [TURN] \oplus [user] \oplus U_t \oplus [system] \oplus S_t$ is the utterance in the $t$-th turn of the dialogue, where $U_t$ and $S_t$ respectively refer to the user's utterance and the system's utterance in the $t$-th turn. We use special tokens $[user]$ and $[sys]$ as prefixes for user utterance sequences and system utterance sequences, respectively. The prefix for the current turn utterance sequence is $[CURR\_TURN]$, and $[TURN]$ separates each turn of dialogue ut-

terance.

**State Sequence**: We convert the dialogue states into natural language sequences by using the heuristic script by DS2 (Shin et al., 2022). Figure 3 shows an illustrative example of using heuristic script to convert dialogue states into natural language sequences. Given a slot-value pair "*(hotel-people, 6)*", since the slot value is not "*dontcare*", the slot name "*hotel-people*" finds the corresponding fragment "*for people*" from the slot sequence fragment. By using the slot value "*6*" to fill in the missing part of the sequence, we obtain "*for 6 people*", then add the commmon sequence fragment "*The user is looking for a place to stay*". The final sequence form is obtained as "*The user is looking for a place to stay for 6 people*". Specifically, when the slot value is "*dontcare*", the slot name needs to be searched for the corresponding fragment from the Dontcare slot sequence fragment. Moreover, we define the dialogue state at turn $t$ as $B_t = \{(S_j, V_j^t) \mid 1 \le j \le J\}$, where $V_j^t$ is the cor-

responding value of the slot $S_j$, and $J$ represents the size of a set of predefined slots. $S_j$ refers to the domain and slot names connected by "-". $B_t^{pred}$ in Figure 2 represents the predicted dialogue states for the $t$-th turn, while $B_t^{pred\_seq}$ refers to the sequence obtained by converting $B_t^{pred}$. the goal of Scalable-DSC is to correct the errors in *State Sequence* $B_t^{pred}$, while STP-DST is to generate the $B_t^{pred}$ based on the previous dialogue state (*State Sequence*) $B_{t-1}^{pred}$ and the dialogue history $H_t$.

**Template Options**: We concatenates all the sequence fragments in order (*Common sequence fragment* and *Slot sequence fragment*) to obtain it, providing the model with more granular guidance. We replace the missing slot value portions in the fragments using special tokens "$[value\_i]$"($i = 1, ..., J$). It is worth noting that we have designed slot values containing "*any*" as a replacement for the "*dontcare*" slot value [1]. The benefit of this is that template options do not require connecting *Dontcare slot sequence fragment* of each slot, reducing the input length of the template sequence. In Figure 2, $T_{DSC}$ and $T_{DST}$ respectively represent the template options for Scalable-DSC and STP-DST. Especially, for Scalable-DSC, $T_{DSC}$ added a sequence at the beginning of $T_{DST}$: "*(The following dialogue status is incorrect | Actually)*". This sequence serves as a guide to instruct Scalable-DSC in inferring incorrect dialogue state sequence and then correcting them.

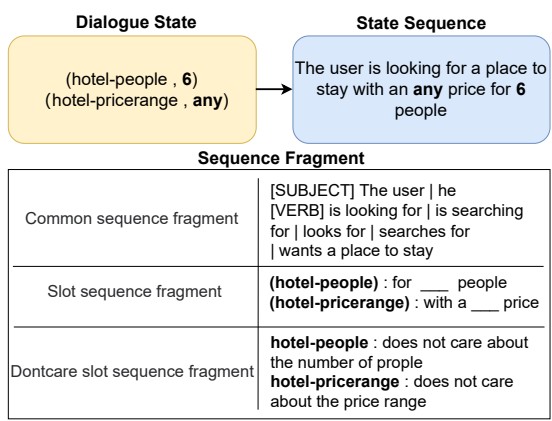

Figure 3: The process of converting dialogue states into sequence of dialogue state transitions. "__" represents the position for filling slot values using the regularization approach.

**Slot Options**: We establish the connection between the detailed description of the $i$-th slot and

[1]The specific replacement slot values for each slot are shown in Appendix C.

the $i$-th masked slot value position in the template options by employing the same special token "$[value\_i]$"($i = 1, ..., J$). The advantage of this lies in that when the model needs to fill in the slot values masked by special tokens in the template, it is able to associate with the corresponding slot's detailed description information through the special tokens. This helps the model comprehend the semantics of the masked slot value position and enables more accurate text completion. The slot options for all slots is represented as $O = [value\_1] : desc_1 \oplus \cdots \oplus [value\_J] : desc_J$, where $desc_i$ is the connection between the domain name of the $i$-th slot and the detailed description of the $i$-th slot. For example, the detailed description of the slot named "*restaurant-food*" is: "*restaurant-the cuisine of the restaurant you are looking for*".

## 2.2 Sequence Generation in STP Approach

The model automatically selects relevant template fragments from the template options and replaces the special token "[value_$i$]" in the fragments with actual slot values to generate a corrected dialogue state natural language sequence. Especially, the generation process of Scalable-DSC can be divided into two steps: (1) Generating the phrase "*The following dialogue status is incorrect*" and then generate the incorrect parts of the state sequence; (2) Generating the phrase "*Actually*" and then generating a complete and correct state sequence. As shown in Figure 2, we define the erroneous part of the state sequence as $B_t^{error\_seq}$ and the correct state sequence as $B_t^{correct\_seq}$. It is worth noting that when there are no errors in the state sequence, Scalable-DSC directly generate the sequence $B_t^{correct\_seq}$.

## 2.3 Training Strategy of Scalable-DSC

We propose two-stage training of the Scalable-DSC model, with two training strategies for the stages. Strategy 1, we replicate the predictive state simulator algorithm for Correctable-DST (Xie et al., 2022) (Refer to Appendix A for more details). Using this algorithm, we randomly replace slot values in dialogue state labels based on a certain probability $\beta$ to simulate predicted dialogue states $B_t^{simulate}$ for the $t$-th turn, which may contain wrong slot values, as training data to train Scalable-DSC. Strategy 2, We use the DST model to predict the dialogue state of the training set as the training set for Scalable-DSC.

## 2.4 Training Objective

The model is trained with a maximum likelihood objective function. Given a training sample $d = \{X, Y\}$, where the $X$ is the encoder context of the model, and $Y$ is the target output text of the model. The encoder context of STP-DST is defined as $X_{DST}$ = *Dialogue History* : $\oplus H_t \oplus$*State Sequence* : $\oplus B_{t-1}^{pred\_seq} \oplus$*Slot Options* : $\oplus O \oplus$ *Template options* : $\oplus T_{DST}$ and $Y_{DST}$ = $B_t^{pred\_seq}$ is the target output text of STP-DST. In the Scalable-DSC model, the encoder context is defined as $X_{DSC}$ = *Dialogue History* : $\oplus D_t \oplus$*State Sequence* : $\oplus B_t^{pred\_seq} \oplus$*Slot Options* : $\oplus O \oplus$ *Template Options* : $\oplus T_{DSC}$ and the target output text $Y_{DSC} = B_t^{error\_seq} \oplus B_t^{correct\_seq}$. The objective of model $\mathcal{L}_\Theta$ is defined as

$$\mathcal{L}_\Theta = -\sum_{i=1}^{|Y|} \log P_\Theta(Y_i | Y_{<i}; X), \quad (1)$$

where $\Theta$ represents the parameters of the model.

## 3 Experiments

### 3.1 Dataset

We adopt MultiWOZ 2.0-2.4(Budzianowski et al., 2018; Eric et al., 2020; Zang et al., 2020; Han et al., 2021; Ye et al., 2021a) as the datasets in our experiments. MultiWOZ is a multi-domain dialogue dataset that consists of dialogues between simulated users and dialogue systems. The training, validation and test sets in the MultiWOZ dataset contain 8420, 1000 and 1000 dialogues respectively. The model we follow the previous work (Wu et al., 2019; Lee et al., 2019; Kim et al., 2020; Le et al., 2020; Ye et al., 2021b; Wang et al., 2022) in training and only used data from five domains (restaurant, hotel, attraction, taxi, train).

### 3.2 Evaluation Metric

We borrow and modify the DS2 (Shin et al., 2022) heuristic script to convert the generated state sequences of the model into dialogue states for evaluation. The model is automatically evaluated based on two metrics (Williams et al., 2013). **Joint Goal Accuracy** is the primary evaluation metric for cross-domain DST tasks. It is the percentage of turns for which all the slots are correctly identified. **Slot Accuracy** only considers individual slot-level accuracy, calculating the proportion of the correct slot values filled in the slot from a macro perspective. **Final Joint Goal Accuracy** Xie et al. (2022)

is defined as the proportion of examples (dialogues) where the predicted dialogue state of last turn exactly matches the ground-truth dialogue state of the last turn.

### 3.3 Settings

We initialize the Scalable-DSC and STP-DST model parameters with pre-trained T5-base (Raffel et al., 2020) from Huggingface Transformers (Wolf et al., 2020). We set the learning rate to 5e-5, and the batch size to 6. The probability $\beta$ is set to 0.06. We optimize the model parameters by using the AdamW optimizer (Loshchilov and Hutter, 2019). The model training was performed on two NVidia RTX 3090 GPUs.

### 3.4 Baselines

We use the following models as the baselines for comparisons: STAR (Ye et al., 2021b) captures slot correlations by a self-attention mechanism; DSS-DST (Guo et al., 2021) proposes a slot status binary classifier to determine whether a slot should update its slot value or inherit its slot value; FPDSC (Zhou et al., 2021) is a multi-level fusion approach that integrates dialogue history and previous dialogue state; LUNA (Wang et al., 2022) is a DST approach for aligning slot and dialogue utterances; DSDN (Xu et al., 2023) dynamically utilizes the information of the previous dialogue state to track the user's goal; Tripy (Heck et al., 2020) proposes three copying approaches for filling slot values; SOM-DST (Kim et al., 2020) is a DST model that utilizes BERT encoding and employs an RNN copying mechanism for decoding dialogue states; MinTL (Lin et al., 2020) is a DST model that predicts only a subset of dialogue states; TransformerDST (Zeng and Nie, 2020) proposes using only BERT as both the encoder and decoder; SimpleTOD (Hosseini-Asl et al., 2020) is an end-to-end dialogue system that formulates dialogue state tasks as sequential tasks; DS2 (Shin et al., 2022) converts dialogue states into sequence texts to track user's goals; AG-DST (Tian et al., 2021) proposes the use of secondary decoding to achieve the goal of correcting the dialogue state; D3ST (Zhao et al., 2022) proposes the approach of combining schema information and intent information to accomplish the dialogue state tracking; Correctable-DST (Xie et al., 2022) proposes to first obtain incorrect slot information in the dialogue state and use these incorrect slot information to guide the correction of the DST model in generating the correct dialogue

| Model | MultiWOZ(%) | | | | |
|---|---|---|---|---|---|
| | 2.0 | 2.1 | 2.2 | 2.3 | 2.4 |
| **Predefined ontology** | | | | | |
| DSS-DST | **56.93** | **60.73** | 58.04 | - | - |
| FPDSC | 53.17 | 59.07 | - | - | - |
| LUNA | 55.31 | 57.62 | 56.13 | - | - |
| DSDN | 55.30 | 59.25 | - | - | - |
| **Open-vocabulary** | | | | | |
| TripPy | - | 55.29 | - | 63.0♠ | 59.62◇ |
| MinTL | 51.41*/52.1 | 52.92*/53.62 | 55.82* | 56.95* | 67.56* |
| TransformerDST | 53.71*/54.64 | 55.56*/55.35 | 55.64* | 57.17* | 69.74* |
| SimpleTOD | - | 55.76 | 54.02‡ | 51.3♠ | - |
| DS2 | 54.78 | 52.32 | - | - | - |
| D3ST | - | 57.8 | 58.7 | 60.8 | 75.9 |
| AG-DST | - | - | 57.26‡ | - | - |
| Correctable-DST | **67.51** | **68.24** | **70.30** | **71.38** | **81.27** |
| **SOM-DST** | 52.61*/51.38 | 52.47*/52.57 | 53.27* | 55.69*/55.5♠ | 67.54*/66.78◇ |
| +Scalable-DSC[N] | 65.56(0.06) | 65.81(0.06) | 67.06(0.06) | 67.11(0.06) | 74.13(0.06) |
| +Scalable-DSC[R] | 66.13 | 65.32 | 66.42 | 67.20 | 73.80 |
| +Scalable-DSC[R+N] | 65.41 | **67.02** | **68.56** | **67.39** | **74.92** |
| +Scalable-DSC[N+R] | **68.61** | 66.83 | 67.14 | 67.23 | 74.37 |
| **STAR** | 52.26*/54.53 | 54.08*/56.36 | 60.49* | 65.87* | 74.63*/73.62◇ |
| +Scalable-DSC[N] | 67.94(0.06) | 68.94(0.06) | 71.47(0.06) | 72.62(0.06) | 79.60(0.06) |
| +Scalable-DSC[R] | 64.54 | 68.44 | 71.07 | 72.45 | 78.60 |
| +Scalable-DSC[R+N] | **68.92** | 67.85 | **71.63** | 71.74 | 77.81 |
| +Scalable-DSC[N+R] | 68.51 | **69.02** | 71.33 | **72.50** | **78.91** |
| **STP-DST** | 55.51 | 55.85 | 57.08 | 63.79 | 76.31 |
| +Scalable-DSC[N] | 65.73(0.06) | 68.04(0.06) | 71.11(0.06) | 74.47(0.06) | 81.17(0.06) |
| +Scalable-DSC[R] | 65.97 | 67.76 | 71.31 | 74.56 | 81.40 |
| +Scalable-DSC[R+N] | **67.91** | 68.35 | 71.64 | **74.92** | 80.52 |
| +Scalable-DSC[N+R] | 66.10 | **69.13** | **71.83** | 74.46 | **81.62** |

Table 1: Joint goal accuracy (%) on the test set of MultiWOZ2.0-2.4. ⋆ the results borrowed from Xie et al. (2022). ♠: the results borrowed from Zang et al. (2020). ‡: the results borrowed from Tian et al. (2021). ♠: results are cited from the 2.3 websites https://github.com/lexmen 318/MultiWOz-coref. ◇: the results borrowed from (Ye et al., 2021a). "-" indicates no public number is available. "(0.06)" refers to the predictive state simulator's probability.

state. Upper part of Table 1 shows the baseline results.

## 4 Experiment Results

### 4.1 Main Results

The lower part of Table 1 shows the results when applying Scalable-DSC with different training configurations to correct the predicted dialogue states of three DST models(STP-DST, SOM-DST(Kim et al., 2020) and STAR(Ye et al., 2021b)). As can be seen, Scalable-DSC correct the erroneous dialogue states predicted by STP-DST and achieves state-of-the-art performance on MultiWOZ 2.0-2.4 with joint goal accuracy of 67.91%, 69.13%, 71.83%, 74.92%, and 81.62%. Furthermore, the dialogue states containing erroneous slot values predicted by the SOM-DST and STAR models can also be corrected by Scalable-DSC. Experimental results have demonstrated that the joint target accuracy of these two models is significantly improved after correction by Scalable-DSC. This shows that Scalable-DSC is able to effectively correct erroneous dialogue state predicted by any DST model, demonstrating its scalability. Furthermore, to investigate the impact of different training strategies on the corrective performance of Scalable-DSC, we evaluate four training configurations: (1) [N]: Training Scalable-DSC using predicted dialogue states simulated by a predictive state simulator as the training data. (2) [R]: Training Scalable-DSC using predicted dialogue states from DST as the training data. (3) [N+R]: First using [N] strategy and then the [R] strategy to train the Scalable-DSC model in stages. (4) [R+N]: First using the [R] strategy and then the [N] strategy to train the Scalable-DSC model in stages. We can see from Table 1 that

| Model | Joint Goal Accuracy(%) |
|---|---|
| Scalable-DSC[N] | **81.17** |
| - *any* | 79.91(-1.26) |
| - *slot options* | 80.23(-0.94) |
| - *template options* | 79.20(-1.97) |
| Scalable-DSC[R] | **81.40** |
| - *any* | 80.49(-0.91) |
| - *slot options* | 81.17(-0.23) |
| - *template options* | 80.41(-0.99) |

Table 2: Ablation analysis of different prompts for STP on the MultiWOZ2.4 test set.

the correction performance of training Scalable-DSC in stages, combining two strategies, exceeds the performance of training Scalable-DSC using a single strategy only.

# 5 Analysis

## 5.1 Ablation Analysis of STP Approach.

We conduct ablation studies to investigate into the impact of different prompts in STP. Table 2 shows the results on Scalable-DSC trained with different training strategies to correct erroneous dialogue states predicted by STP-DST. The results show that when the slot value "dontcare" is not replaced with "any", the joint goal accuracy of Scalable-DSC trained separately using two strategies is reduced by 1.26% and 0.91% respectively after correction. We believe that the reason for the performance decline is that the template options require connecting the *Dontcare slot sequence fragment* of each slot. This not only increases the input length of the model but also requires additional decoding of the dontcare sequence, adding pressure to the decoding process of model. The inputs of Scalable-DSC trained separately using two training strategies do not concatenate slot options. Their performance decreases by 0.94% and 0.23% respectively. This demonstrates that with the assistance of slot options, the model is capable of gaining a better understanding of the implied meanings of masked slot values within template options. Furthermore, when the model is trained without template options as the prompt texts, the performance of the two training strategies decreased by 1.97% and 0.99% respectively. These results elucidate the effectiveness of template options in enabling the model to generate controlled corrective information, thereby enhancing its corrective performance.

| MultiWOZ | Model | Slot Acc↑(%) | O↓ | P↓ | E↓ |
|---|---|---|---|---|---|
| | | | **Slot Error↓(%)** | | |
| 2.0 | STP-DST | 97.56 | 0.66 | 1.39 | 0.39 |
| | +Scalable-DSC[N] | 98.32 | **0.06** | 1.22 | 0.40 |
| | +Scalable-DSC[R] | 98.33 | 0.10 | 1.17 | 0.40 |
| | +Scalable-DSC[R+N] | **98.49** | 0.10 | **1.00** | 0.41 |
| | +Scalable-DSC[N+R] | 98.36 | 0.10 | 1.14 | **0.40** |
| 2.1 | STP-DST | 97.58 | 0.78 | 1.26 | 0.38 |
| | +Scalable-DSC[N] | 98.35 | 0.02 | 1.26 | 0.37 |
| | +Scalable-DSC[R] | 98.44 | 0.02 | 1.17 | 0.37 |
| | +Scalable-DSC[R+N] | 98.46 | **0.01** | 1.29 | **0.24** |
| | +Scalable-DSC[N+R] | **98.54** | 0.04 | **1.04** | 0.38 |
| 2.2 | STP-DST | 97.94 | 0.67 | 0.87 | 0.52 |
| | +Scalable-DSC[N] | 98.68 | 0.01 | 0.82 | 0.49 |
| | +Scalable-DSC[R] | 98.67 | 0.02 | 0.81 | 0.50 |
| | +Scalable-DSC[R+N] | 98.69 | **0.01** | 0.81 | **0.49** |
| | +Scalable-DSC[N+R] | **98.70** | 0.05 | **0.74** | 0.51 |
| 2.3 | STP-DST | 98.30 | 0.34 | 0.70 | 0.66 |
| | +Scalable-DSC[N] | 98.82 | 0.02 | 0.69 | 0.47 |
| | +Scalable-DSC[R] | 98.83 | **0.01** | 0.67 | 0.49 |
| | +Scalable-DSC[R+N] | **98.99** | 0.08 | **0.61** | **0.32** |
| | +Scalable-DSC[N+R] | 98.84 | 0.03 | 0.64 | 0.49 |
| 2.4 | STP-DST | 98.85 | 0.27 | 0.63 | 0.25 |
| | +Scalable-DSC[N] | 99.18 | 0.02 | 0.55 | 0.25 |
| | +Scalable-DSC[R] | 99.25 | **0.01** | 0.50 | **0.24** |
| | +Scalable-DSC[R+N] | 99.20 | 0.03 | 0.52 | 0.25 |
| | +Scalable-DSC[N+R] | **99.27** | 0.01 | **0.48** | **0.24** |

Table 3: Slot Accuracy and Slot Error Rate on MultiWOZ2.0-2.4. **O** refers to over prediction type, **P** refers to partial prediction type, and **E** refers to erroneous prediction type. ↑: higher is better and ↓: lower is better.

## 5.2 Effectiveness Analysis of Scalable-DSC

Analysis on Scalable-DSC is carried out to obtain deeper insights into the reason for the effectiveness. We continue to employ STP-DST as the target correction model, and utilize the Scalable-DSC model to correct its predicted dialogue states. Following (Quan and Xiong, 2020; Xie et al., 2022), we categorize the slot errors into three types: **over** prediction, **partial** prediction, and **erroneous** prediction. Table 3 presents the slot accuracy and the slot error rate of different model combinations on the MultiWOZ 2.0-2.4. As shown, the Scalable-DSC model improves the slot accuracy of STP-DST, demonstrating the benefits brought by correction. It should be noted that the largest proportion of slot type errors in the STP-DST model is lack of predictions, followed by over predictions, and the smallest is updated predictions. After being corrected by Scalable-DSC, we can observe a decrease in the proportions of all three types of slot errors in the STP-DST model. Specifically, we found that Scalable-DSC primarily corrects the error of over prediction in STP-DST. This verifies that the im-

| MultiWOZ | STP-DST | Scalable-DSC[N] | | | Scalable-DSC[R] | | | Scalable-DSC[R+N] | | | Scalable-DSC[N+R] | | |
|---|---|---|---|---|---|---|---|---|---|---|---|---|---|
| | PE↓ | AE↓ | RE↑ | CE↓ | AE↓ | RE↑ | CE↓ | AE↓ | RE↑ | CE↓ | AE↓ | RE↑ | CE↓ |
| **2.0** | 5578 | 153 | 1901 | 3830 | 243 | 2002 | 3819 | 250 | 2374 | 3454 | 199 | 2040 | 3737 |
| **2.1** | 5528 | 60 | 1803 | 3785 | 65 | 2001 | 3592 | 125 | 2090 | 3563 | 97 | 2261 | 3364 |
| **2.2** | 4698 | 36 | 1690 | 3044 | 42 | 1683 | 3057 | 30 | 1701 | 3027 | 123 | 1800 | 3021 |
| **2.3** | 3894 | 10 | 1190 | 2714 | 14 | 1203 | 2705 | 131 | 1315 | 2710 | 77 | 1211 | 2760 |
| **2.4** | 2542 | 70 | 766 | 1864 | 432 | 1319 | 1655 | 91 | 842 | 1791 | 422 | 1336 | 1628 |

Table 4: Error analysis of Scalable-DSC on MultiWOZ 2.0-2.4. "**PE**": the number of incorrectly predicted slot values by the DST model. "**AE**": the number of newly introduced error slots after being corrected by the Scalable-DSC model. "**RE**": the reduced number of error slots after correction by Scalable-DSC. "**CE**": the number of error slots after Scalable-DSC model correction.

provements in the Scalable-DSC model primarily come from correcting over prediction errors. The relevant results of the SOM-DST model and the STAR model are shown in Appendix F. We have also observed, however, that for the other two types of errors (**partial** prediction and **erroneous** prediction), the Scalable-DSC model brings less benefit in terms of correction.

## 5.3 Error Analysis of Scalable-DSC

The aforementioned experiments have shown the effectiveness of Scalable-DSC. Here we provide further analysis on the error correction quality to gain more insights into why it could be beneficial. We first investigate whether Scalable-DSC introduced new errors by correcting the dialogue states predicted by STP-DST (i.e., whether Scalable-DSC corrected correct slot values into incorrect ones). To achieve this, we calculate the number of slots correctly predicted by STP-DST but erroneously corrected by Scalable-DSC. The "AE" column in Table 4 displays this quantity. We can observe that Scalable-DSC introduces new slot errors during the correction process, although the number is small. Next, we investigate whether the number of erroneous slot predictions from the DST model decreased due to the correction by the Scalable-DSC model. The results in the "RE" column of Table 4 indicate that the Scalable-DSC model significantly reduces the number of incorrectly predicted slots by correcting erroneous slot values in the STP-DST model. It is worth noting that the number of newly introduced errors is proportional to the number of errors corrected by the model. In other words, the stronger the model's correction ability, the more new errors it will introduce.

## 6 Related Work

Early DST methods(Williams and Young, 2007; Thomson and Young, 2010; Wang and Lemon, 2013b) relied on pre-defined rules to recognize dialogue states. Although good prior knowledge can solve cold-start performance issues, it requires a large amount of manual rule-making, is inflexible, lacks scalability, and cannot simultaneously track multiple types of state information. Subsequently, some classification DST models based on a predefined ontology were proposed (Henderson et al., 2014; Mrkšić et al., 2016; Zhong et al., 2018; Ye et al., 2021b), which significantly improved the model performance. However, it is difficult to design a complete and robust pre-defined ontology file (Xu and Hu, 2018). Therefore, most researchers focus on developing open-vocabulary DST models (Gao et al., 2019; Wu et al., 2019; Chen et al., 2020; Shin et al., 2022; Su et al., 2021; Zhao et al., 2022), which extract dialogue states directly from the complete dialogue history, ignoring the issue of incomplete context information caused by truncation of dialogue history. To address this challenge, (Tian et al., 2021; Zhao et al., 2021; Zhou et al., 2021; Xu et al., 2023) attempts to track users' goals from the context of the dialogue and the previous dialogue state. However, error propagation has become the reason for limiting model performance. To address this problem, the idea of correcting dialogue state is proposed (Tian et al., 2021; Xie et al., 2022), but the methods heavily rely on specific DST models, i.e., lacking scalability.

Prompt learning is a widely used technique in natural language processing (NLP), aiming at reducing the gap between the target before training and downstream tasks (Liu et al., 2023). Some recent prompt-based DST work has been proposed(Lee and Jha, 2019; Gao et al., 2020; Lee et al., 2021). They use different text prompts to provide task information to the model. For example, (Su et al., 2021) uses slot names for prompt information, (Rastogi et al., 2020) uses slot description, (Lin et al., 2021) prompt information is possible

values. Some other work (Jiang et al., 2020) has proposed the use of automatic prompts that do not require manually predefined prompts.

## 7 Conclusion and Future Work

We have proposed Scalable-DSC, a new dialogue state correction model aiming at decoupling the error correction functionality from a specific DST model, i.e., scalability, by introducing the STP strategy. The key idea of STP is to convert the dialogue state into a natural language sequence and generates a corrected dialogue state natural language sequence based on the guidance of template options, combined with real dialogue history information. Extensive experiments have been carried out to analyze the scalability and correction performance of the model. Results have confirmed the applicability of the model to other DST models and demonstrated a new state-of-the-art performance on MultiWOZ2.0-2.4. As discussed, limitation still exists. How to better correct all types of errors in dialogue states and avoid the generation of new errors remains interesting research topic for the future work.

## Limitations

This work has two main limitations:

(1) The Scalable-DSC cannot completely correct all types of errors in predicting dialogue states. As mentioned above, the Scalable-DSC model primarily corrects over-prediction errors, while the correction proportion for the other two types of errors is relatively.

(2) The Scalable-DSC has a problem of incorrectly modifying the correct conversation state. From the error analysis experiments, it can be observed that the Scalable-DSC model exhibits a small proportion of incorrect modifications in MultiWOZ2.0-2.4.

## Acknowledgements

We would like to thank the anonymous reviewers for their useful feedback. This work was supported by the Opening Project of Key Laboratory of Xinjiang, China (2020D04047), Natural Science Foundation of China (61663044), the National Key R&D Program of China (2020AAA0107902), and the Excellent Doctoral Student Research Innovation Project of Xinjiang University (No. XJU2023BS065).

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

## A Details of the Predictive State Simulator Algorithm.

The details of simulating predicted dialogue states using the Predictive State Simulator algorithm are presented in Algorithm 1.

---

**Algorithm 1:** Predictive State Simulator

---

**Input** : Dialogue state label $B_t = \{(S_j, V_j^t)\}_{j=1}^{|J|}$;
   Noise probability $\beta$;

1 Create a list of historical values for over prediction $\mathcal{HV}_o$;
2 Create a list of historical values for partial prediction and erroneous prediction $\mathcal{HV}_{lu}$;
3 **for** *slot_id* $j = 1, ..., J$ **do**
4    **if** $V_j^t$ *is not none* **then**
5      | $\mathcal{HV}_o$.append($V_j^t$);
6    **end**
7 **end**
8 $\mathcal{HV}_o$.append($dontcare$);
9 $\mathcal{HV}_{lu} \leftarrow \mathcal{HV}_o$;
10 $\mathcal{HV}_{lu}$.append($none$);
11 **for** *slot_id* $j = 1, ..., J$ **do**
12    Generating random number $beta$;
13    **if** $V_j^t$ *is none* **then**
14      **if** $beta < \beta$ **then**
15        Shuffle $\mathcal{HV}_o$;
16        $V_j^t \leftarrow \mathcal{HV}_o[0]$
17      **end**
18    **else**
19      **if** $beta < \beta$ **then**
20        Shuffle $\mathcal{HV}_{lu}$;
21        $V_j^t \leftarrow \mathcal{HV}_{lu}[0]$
22      **end**
23    **end**
24 **end**

**Output** : Dialogue State $B_t^{noise}$ with Added Noise

---

## B Final joint goal accuracy

| Model | MultiWOZ(%) | | | | |
|---|---|---|---|---|---|
| | 2.0 | 2.1 | 2.2 | 2.3 | 2.4 |
| SOM-DST* | 39.34 | 36.04 | 38.64 | 38.44 | 55.00 |
| MinTL(BartLarge)* | 38.70 | 39.04 | 41.04 | 40.24 | 56.96 |
| Transformer-DST* | 39.69 | 40.27 | 41.54 | 41.30 | 58.40 |
| STAR* | 38.93 | 40.94 | 48.14 | 56.15 | 66.46 |
| Correctable-DST* | **60.36** | **57.86** | **61.66** | **58.26** | **74.17** |
| STP-DST | 39.20 | 40.30 | 41.10 | 52.30 | 66.00 |
| +Scalable-DSC[N] | 53.20 | 54.20 | 59.70 | 61.30 | 71.70 |
| +Scalable-DSC[R] | 53.70 | 55.90 | 58.70 | 61.34 | 75.10 |
| +Scalable-DSC[R+N] | 56.30 | 56.90 | 58.90 | **62.70** | 72.60 |
| +Scalable-DSC[N+R] | 54.10 | **58.32** | 59.10 | 61.23 | **75.50** |

Table 5: Final joint goal accuracy(%) on the MultiWOZ 2.0-2.4 test set. * the results borrowed from Xie et al. (2022).

Table 5 presents the final joint goal accuracy results of our approach on MultiWOZ 2.0-2.4. It is shown that our approach outperforms the previous best results of Correctable-DST on datasets 2.1, 2.2, and 2.4, demonstrating the effectiveness of our method in mitigating the issue of error propagation in dialogues.

## C Slot Value Replacement in Slots

| Slot | Replacement Value |
|---|---|
| hotel-parking, hotel-internet | has any type |
| attraction-type, hotel-type, restaurant-food | any type |
| restaurant-people, hotel-people, hotel-stay, train-people | any num |
| restaurant-time, taxi-leave, taxi-arrive, train-leave, train-arrive | any time |
| restaurant-area, taxi-departure, taxi-destination, train-departure, attraction-area, train-destination, hotel-area, | any where |
| restaurant-pricerange, hotel-stars, hotel-pricerange, restaurant-name, hotel-name, attraction-name | any |
| restaurant-day, hotel-day, train-day | any day |

Figure 4: Replace the "dontcare" in 30 slots with different "any" slot values.

Figure 4 shows the replacement values for the "dontcare" slot values in different slots.

## D Experiment on Per Slot Error Rate

We further investigated the error rates for each slot. We used STP-DST as our primary models and used the Scalable-DSC [N+R] model to correct the predictions of the primary models. The results on the MultiWOZ 2.4 test set are shown in Table 6. It can be seen that the Scalable-DSC model effectively reduces the error rates for most slots. However, we found that slots associated with "name" and "type" have higher error rates in the primary model itself. Although the error rates decrease after correction, these slots still present a challenge.

## E Experiment on the model's generation type.

We believe that when correcting dialogue state, identifying the wrong slot first before correction is more beneficial in reducing the inference burden of the model than direct correction. This idea has been validated as shown in Table 7. On MultiWOZ 2.4, when correcting the predictions of STP-DST, using Scalable-DSC to only generate corrected dialogue state sequences achieved a 80.90% Joint Goal Accuracy (JGA), while using the method proposed

| Slot | STP-DST | +Scalable-DSC[N+R] |
|---|---|---|
| taxi-departure | 1.38 | 0.70 |
| taxi-destination | 1.20 | 0.69 |
| taxi-leave | 0.42 | 0.37 |
| taxi-arrive | 0.55 | 0.32 |
| restaurant-name | 2.02 | 1.46 |
| restaurant-area | 0.75 | 0.83 |
| restaurant-pricerange | 1.26 | 1.03 |
| restaurant-food | 1.15 | 0.84 |
| restaurant-day | 0.24 | 0.21 |
| restaurant-people | 0.37 | 0.26 |
| restaurant-time | 0.47 | 0.40 |
| attraction-name | 2.46 | 1.54 |
| attraction-area | 1.13 | 1.13 |
| attraction-type | 1.83 | 1.73 |
| hotel-name | 1.34 | 0.94 |
| hotel-internet | 1.81 | 1.68 |
| hotel-area | 1.43 | 1.11 |
| hotel-parking | 1.28 | 1.12 |
| hotel-stars | 1.00 | 0.86 |
| hotel-type | 3.73 | 2.12 |
| hotel-pricerange | 1.68 | 1.55 |
| hotel-day | 0.17 | 0.01 |
| hotel-people | 0.37 | 0.08 |
| hotel-stay | 0.17 | 0.05 |
| train-departure | 0.63 | 0.36 |
| train-day | 0.32 | 0.20 |
| train-arrive | 0.74 | 0.66 |
| train-destination | 0.99 | 0.71 |
| train-leave | 2.04 | 1.93 |
| train-people | 1.39 | 0.28 |

Table 6: The error rate of each slot on MultiWOZ 2.4.

in this paper achieved an 81.62% JGA. The experimental results show that our proposed method achieved a 0.72% improvement, proving that identifying the wrong slot first before correction is more beneficial in reducing the inference burden of the model than direct correction.

| Model | JGA(%) | Generation Type |
|---|---|---|
| STP-DST | 76.31 | - |
| +Scalable-DSC[N+R] | 81.62 | Ours |
| +Scalable-DSC[N+R] | 80.90 | Generate corrected states |

Table 7: Joint goal accuracy (%) on the test set of Multi-WOZ2.4.

# F  Experiment on Slot Error Rate

We used SOM-DST and STAR as target correction models, and Table 8 presents the performance after using the Scalable-DSC model to correct the predicted dialogue state of the two DST models on the MultiWOZ 2.0-2.4 dataset. The results have demonstrated that the Scalable-DSC model primarily corrects the over prediction errors of the DST model, which has become the main reason for the strong correction capability of the Scalable-DSC model.

| MultiWOZ | Model | Slot Acc↑(%) | Slot Error Rate↓(%) | | |
|---|---|---|---|---|---|
| | | | Over↓ | lack↓ | Update↓ |
| | SOM-DST | 97.27 | 0.99 | 1.22 | 0.52 |
| | + Scalable-DSC[N] | 98.39 | **0.07** | 0.98 | 0.56 |
| | + Scalable-DSC[R] | 98.24 | 0.08 | 1.18 | 0.49 |
| | + Scalable-DSC[R+N] | 98.43 | 0.26 | **0.86** | **0.45** |
| 2.0 | + Scalable-DSC[N+R] | **98.45** | 0.10 | 0.91 | 0.55 |
| | STAR | 97.58 | 0.83 | 0.92 | 0.39 |
| | + Scalable-DSC[N] | 98.38 | 0.37 | 0.99 | 0.26 |
| | + Scalable-DSC[R] | 98.16 | 0.39 | 1.02 | 0.43 |
| | + Scalable-DSC[R+N] | **98.49** | 0.26 | **0.85** | 0.40 |
| | + Scalable-DSC[N+R] | 98.48 | **0.24** | 0.98 | **0.30** |
| | SOM-DST | 97.37 | 0.90 | 1.14 | 0.59 |
| | + Scalable-DSC[N] | 98.10 | 0.25 | 1.09 | 0.56 |
| | + Scalable-DSC[R] | 98.23 | 0.14 | 1.08 | 0.55 |
| | + Scalable-DSC[R+N] | **98.33** | **0.05** | 1.02 | 0.59 |
| 2.1 | + Scalable-DSC[N+R] | 98.32 | 0.19 | **0.94** | **0.51** |
| | STAR | 97.36 | 0.78 | 1.46 | 0.40 |
| | + Scalable-DSC[N] | 98.72 | 0.11 | **0.81** | **0.35** |
| | + Scalable-DSC[R] | 98.64 | 0.12 | 0.86 | 0.38 |
| | + Scalable-DSC[R+N] | 98.69 | 0.10 | 0.84 | 0.37 |
| | + Scalable-DSC[N+R] | **98.75** | **0.02** | 0.84 | 0.39 |
| | SOM-DST | 97.47 | 0.85 | 1.06 | 0.62 |
| | + Scalable-DSC[N] | 98.35 | 0.11 | 0.98 | **0.56** |
| | + Scalable-DSC[R] | 98.29 | 0.15 | 0.96 | 0.60 |
| | + Scalable-DSC[R+N] | **98.40** | **0.04** | 0.95 | 0.62 |
| 2.2 | + Scalable-DSC[N+R] | 98.33 | 0.14 | 0.96 | 0.57 |
| | STAR | 97.93 | 0.69 | 0.98 | 0.4 |
| | + Scalable-DSC[N] | 98.61 | 0.18 | 0.83 | **0.38** |
| | + Scalable-DSC[R] | 98.59 | 0.06 | 0.96 | 0.39 |
| | + Scalable-DSC[R+N] | **98.74** | **0.03** | 0.83 | 0.39 |
| | + Scalable-DSC[N+R] | 98.62 | 0.19 | **0.81** | 0.38 |
| | SOM-DST | 97.64 | 0.61 | 0.83 | 0.93 |
| | + Scalable-DSC[N] | 98.37 | 0.05 | 0.82 | 0.76 |
| | + Scalable-DSC[R] | 98.38 | **0.01** | 0.84 | **0.77** |
| | + Scalable-DSC[R+N] | **98.41** | 0.02 | **0.79** | 0.78 |
| 2.3 | + Scalable-DSC[N+R] | 98.38 | 0.01 | 0.84 | 0.77 |
| | STAR | 98.13 | 0.46 | 0.70 | 0.72 |
| | + Scalable-DSC[N] | 98.67 | 0.01 | 0.74 | 0.58 |
| | + Scalable-DSC[R] | 98.65 | 0.02 | 0.74 | 0.59 |
| | + Scalable-DSC[R+N] | 98.48 | 0.25 | **0.64** | 0.63 |
| | + Scalable-DSC[N+R] | **98.69** | **0.01** | 0.73 | **0.57** |
| | SOM-DST | 98.58 | 0.31 | 0.67 | 0.44 |
| | + Scalable-DSC[N] | 98.76 | 0.20 | 0.73 | 0.30 |
| | + Scalable-DSC[R] | 98.86 | 0.11 | 0.61 | **0.43** |
| | + Scalable-DSC[R+N] | **98.96** | 0.04 | **0.57** | 0.44 |
| 2.4 | + Scalable-DSC[N+R] | 98.90 | **0.04** | 0.63 | 0.43 |
| | STAR | 98.96 | 0.20 | 0.62 | **0.21** |
| | + Scalable-DSC[N] | 99.11 | 0.09 | 0.57 | 0.23 |
| | + Scalable-DSC[R] | 99.10 | 0.10 | 0.57 | 0.23 |
| | + Scalable-DSC[R+N] | 99.10 | 0.11 | 0.57 | 0.22 |
| | + Scalable-DSC[N+R] | **99.23** | **0.06** | **0.49** | 0.22 |

Table 8: Slot Accuracy and Slot Error Rate on MultiWOZ2.0-2.4. ↑: higher is better and ↓: lower is better.