# OpenReview forum: "Scalable-DSC: A Structural Template Prompt Approach to Scalable Dialogue State Correction"
_EMNLP/2023/Conference — EMNLP 2023 Main_

### Official Review · Reviewer_QXpz · 2023-07-27

**Soundness:** 4

**Excitement:**

3: Ambivalent: It has merits (e.g., it reports state-of-the-art results, the idea is nice), but there are key weaknesses (e.g., it describes incremental work), and it can significantly benefit from another round of revision. However, I won't object to accepting it if my co-reviewers champion it.

**Paper Topic And Main Contributions:**

State of the art methods for dialogue state tracking (DST) sequentially predict the state (ie a set of slots and their values) based on the dialogue history and the previous state.  One problem with this approach is that errors in state prediction tend to propagate forward compounding the error rate.  A recent paper by Xie et al (Correctable-DST) showed that incorporating an explicit slot error detector can reduce this problem and their results represent the current SOTA on the much used MultiWoz dataset.

This paper attempts to provide a similar slot error correction facility whilst treating the dialogue state tracker as a black box.  To do this they train a T5 based transformer to take as input the DST output and generate an output which explicitly identifies errors.  The input prompt is augmented with a set of manually crafted the slot options and templates describing the possible output forms.

The system is trained on MultiWoz augmented by induced slot errors generated by a either a so-called 'Predictive State Simulator' or by the DST itself.  The results are comparable or slightly better than the earlier Correctable-DST.

**Questions For The Authors:**

Why do you structure the Scalable-DSC output in the form of error + correction?  Is it not possible to train the model to simply output the corrected state description?


**Reasons To Accept:**

The proposed system is shown to reduce slot error propagation and overall results in terms of joint accuracy match or exceed the SOTA.  Additionally, the method can be used with existing DSTs as a post-processor therefore making it more widely applicable.  It is tested using 3 different DSTs and it substantially improves each of them.


**Reasons To Reject:**

The method requires slot options to be enumerated and templates written to describe DST outputs and corrections in natural language.  The system is tested on just one data set so it is unproven as to the general usefulness of this approach.  The overall results match the SOTA but do not substantially improve it.


**Reproducibility:**

3: Could reproduce the results with some difficulty. The settings of parameters are underspecified or subjectively determined; the training/evaluation data are not widely available.

**Reviewer Confidence:**

4: Quite sure. I tried to check the important points carefully. It's unlikely, though conceivable, that I missed something that should affect my ratings.

**Typos Grammar Style And Presentation Improvements:**

line 200 - is the reference to Fig 3 correct?

---

> ### Author Rebuttal · Authors · 2023-08-29
>
> We feel great thanks for your professional review work on our article. As you are concerned, there are several problems that need to be addressed. The responses to the comments are listed as follows.
>
> ### **Questions:**
>
> **Q1: Why do you structure the Scalable-DSC output in the form of error + correction? Is it not possible to train the model to simply output the corrected state description?**
>
> | Model              | MultiWOZ 2.4 | Generation Type                                     |
> | ------------------ | :----------: | --------------------------------------------------- |
> | STP-DST            |    76.31     | -                                                   |
> | +Scalable-DSC[N+R] |    81.62     | generate erroneous states+generate corrected states |
> | +Scalable-DSC[N+R] |    80.90     | generate corrected states                           |
>
> **Table 1.** Joint goal accuracy (%) on the test set of MultiWOZ2.4. "[N+R]": First using [N] strategy and then the [R] strategy to train the Scalable-DSC model in stages. (All parameter settings for this experiment are consistent with those mentioned in our paper. ）
>
> **Response1:**
>
> We believe that when correcting dialogue state, identifying the wrong slot first before correction is more beneficial in reducing the inference burden of the model than direct correction. This idea has been validated as shown in Table 2. On the MultiWOZ 2.4 dataset, when correcting the predictions of the STP-DST model, using Scalable-DSC to only generate corrected dialogue state sequences achieved a 80.90% Joint Goal Accuracy (JGA), while using the method proposed in this paper achieved an 81.62% JGA. The experimental results show that our proposed method achieved a 0.72% improvement, proving that identifying the wrong slot first before correction is more beneficial in reducing the inference burden of the model than direct correction.  These will be added in the improved manuscript along with some explanation and analysis.
>
> ### **Reasons To Reject:**
>
> **Q1:  The method requires slot options to be enumerated and templates written to describe DST outputs and corrections in natural language. The system is tested on just one data set so it is unproven as to the general usefulness of this approach. The overall results match the SOTA but do not substantially improve it.**
>
> | model              | MultiWOZ 2.0  | MultiWOZ 2.1  | MultiWOZ 2.2  | MultiWOZ 2.3  | MultiWOZ 2.4 |
> | ------------------ | :-----------: | :-----------: | :-----------: | :-----------: | :----------: |
> | SOM-DST            |     51.38     |     52.57     |     53.27     |     55.50     |    66.78     |
> | +Scalable-DSC[N+R] | 68.61(+17.23) | 66.83(+14.26) | 67.14(+13.87) | 67.23(+11.73) | 74.37(+7.59) |
> | STAR               |     54.53     |     56.36     |     60.49     |     65.87     |    73.62     |
> | +Scalable-DSC[N+R] | 68.51(+13.98) | 69.02(+12.66) | 71.33(+10.84) | 72.50(+6.63)  | 78.91(+5.29) |
> | STP-DST            |     55.51     |     55.85     |     57.08     |     63.79     |    76.31     |
> | +Scalable-DSC[N+R] | 66.10(+10.59) | 69.13(+13.28) | 71.83(+14.75) | 74.46(+10.67) | 81.62(+5.31) |
>
> **Table2.**  Joint goal accuracy (%) on the test set of MultiWOZ2.4. "[N+R]": First using [N] strategy and then the [R] strategy to train the Scalable-DSC model in stages. (All parameter settings for this experiment are consistent with those mentioned in our paper. ）
>
> **Response1:**
>
> The publicly available datasets used for DST evaluation include DSTC2, Wizard-of-OZ, Schema-Guided Dialogue dataset, and MultiWOZ 2.0-2.4. We believe that this method can also be applied to other datasets beyond MultiWOZ 2.0-2.4.  The structural templates we constructed were generated using a heuristic converter that supports interconversion between dialogue states and natural language sequences, and the heuristic converter we used was extracted and refactored from the work of DS2 [1]. The Appendix A4 section in DS2 describes how to extend the template generation approach to new domain data, It explains that the developer only needs to rewrite the dialogue heuristic converter between states and summaries is sufficient, similarly extending our proposed method to new domain data requires only a slight modification of the code for the heuristic converter, which can be implemented by a Python developer in just a few hours, ensuring that our correction method is easily scalable to other datasets. For a more comprehensive validation, we will release the code to replication or potentially supplementing experiments on GitHub in the near future.
>
> We believe that the main contribution of this paper is that the Scalable-DSC model can correct prediction errors made by other DST models. Additionally, the performance improvement achieved by the Scalable-DSC model surpasses that of the Correctable-DST model, as shown in Table 2. On the MultiWOZ 2.0-2.4 dataset, the performance enhancement brought by the Scalable-DSC model to three DST models ranges between 5.31% and 17.23%, demonstrating a significant improvement.
>
> ### **Typos Grammar Style And Presentation Improvements:**
>
> **Q1: line 200 - is the reference to Fig 3 correct?**
>
> **Response1:**
>
> Thank you for pointing out our writing error. It will help us improve the quality of the manuscript. This should be Figure 2, and we will correct this mistake.
>
> We thank the reviewers for pointing out our problems and helping us to revise the manuscript. We have considered all suggestions and will correct errors in an improved manuscript.
>
> ### **References**
>
> [1] Shin J, Yu H, Moon H, et al. Dialogue Summaries as Dialogue States (DS2), Template-Guided Summarization for Few-shot Dialogue State Tracking[C]//Findings of the Association for Computational Linguistics: ACL 2022. 2022: 3824-3846.
>
> [2] Xie H, Su H, Song S, et al. Correctable-DST: Mitigating Historical Context Mismatch between Training and Inference for Improved Dialogue State Tracking[C]//Proceedings of the 2022 Conference on Empirical Methods in Natural Language Processing. 2022: 876-889.

---

### Official Review · Reviewer_5vDD · 2023-08-04

**Soundness:** 3

**Excitement:**

3: Ambivalent: It has merits (e.g., it reports state-of-the-art results, the idea is nice), but there are key weaknesses (e.g., it describes incremental work), and it can significantly benefit from another round of revision. However, I won't object to accepting it if my co-reviewers champion it.

**Paper Topic And Main Contributions:**

This paper is to address the error accumulation problem in dialogue state tracking (DST) task. In this work,  a standalone dialogue state correction model is proposed to help several DST models to handle the errors in prediction. The experimental results show encouraging results.

**Questions For The Authors:**

- What is final joint goal accuracy? It seems no explanations about how to calculate it.

**Reasons To Accept:**

- The motivation is clear. Error accumulation is an important topic to DST task.
- The experimental results somehow show the effectiveness of the proposed approaches.

**Reasons To Reject:**

- I'd like to see what types of the errors are still challenging and I think this part is more important to have a deep understanding about this problem and the proposed approach. It is still not clear how the proposed approaches work to handle what types of errors, e.g., capturing ontologies like hotel names, or selecting options (day of week) from a limited set, or reference error.
- The dataset is too single. The series of MultiWOZ have a very small set of domains (5 domains if the used subset is consistent to previous works) and it contain too many repeated dialogues. I think experiments on more datasets will be more helpful to appeal this work, e.g., Schema Guided Dialogue (SGD) dataset.
- Some details of experimental setup are missing. MultiWOZ 2.2 is a little different to other MultiWOZ series datasets. It add more additional slots. Please clarify if the used slots are same to other datasets or including these additional slots. This part is quite important but it is missing in the paper.
- Lacking analysis. Could you please provide some analysis about what type of errors are improved? At least quantitive analysis like error rate per domain/slot, case studies are helpful to have a better understanding.

**Reproducibility:**

3: Could reproduce the results with some difficulty. The settings of parameters are underspecified or subjectively determined; the training/evaluation data are not widely available.

**Reviewer Confidence:**

5: Positive that my evaluation is correct. I read the paper very carefully and I am very familiar with related work.

---

> ### Author Rebuttal · Authors · 2023-08-29
>
> We feel great thanks for your professional review work on our article. As you are concerned, there are several problems that need to be addressed. The responses to the comments are listed as follows.
>
> ### **Questions:**
>
> **Q1: What is final joint goal accuracy? It seems no explanations about how to calculate it.**
>
> **Response1:**
>
> Thank you for pointing out the issues in the paper. It was indeed our oversight not to introduce and reference the concept of the final joint target accuracy. It is defined the proportion of examples (dialogues) where the predicted dialogue state of last turn exactly matches  the ground-truth dialogue state of last turn. This evaluation metric was proposed by Xie et al[1]. We will clarify this point in the revised manuscript.
>
> ### **Reasons To Reject:**
>
> **Q1: I'd like to see what types of the errors are still challenging and I think this part is more important to have a deep understanding about this problem and the proposed approach. It is still not clear how the proposed approaches work to handle what types of errors, e.g., capturing ontologies like hotel names, or selecting options (day of week) from a limited set, or reference error.**
>
> **Response1:**
>
> In our method, we have explained (1) using a predictive state simulator during the data processing stage to generate dialogue states containing three types of errors; (2) generating predicted dialogue states from other DST models, and during the training phase, enabling our model to learn how to assess and rectify errors in the dialogue states predicted by other models based on the dialogue history. The Scalable-DSC capturing the correct slot values from the dialogue history rather than selecting from a limited set.
>
> **Q2: The dataset is too single. The series of MultiWOZ have a very small set of domains (5 domains if the used subset is consistent to previous works) and it contain too many repeated dialogues. I think experiments on more datasets will be more helpful to appeal this work, e.g., Schema Guided Dialogue (SGD) dataset.**
>
> **Response2:**
>
> The publicly available datasets for DST evaluation include DSTC2, Wizard-of-OZ, Schema-Guided Dialogue datasets and MultiWOZ 2.0-2.4. Thank you very much for your suggestions. We believe that our approach can also be applied to datasets other than MultiWOZ 2.0-2.4.  For a more comprehensive validation, we will release the code to replication or potentially supplementing experiments on GitHub in the near future.
>
> **Q3: Some details of experimental setup are missing. MultiWOZ 2.2 is a little different to other MultiWOZ series datasets. It add more additional slots. Please clarify if the used slots are same to other datasets or including these additional slots. This part is quite important but it is missing in the paper.**
>
> **Response3:**
>
> In lines 306-310 of the paper, we mentioned the utilization of slot information from five domains (restaurant, hotel, attraction, taxi, train) in all versions of the MultiWOZ dataset. Actually, The slots for these five domains remain consistent across the MultiWOZ 2.2 dataset and other datasets.
>
> **Q4: Lacking analysis. Could you please provide some analysis about what type of errors are improved? At least quantitive analysis like error rate per domain/slot, case studies are helpful to have a better understanding.**
>
> | Slot                  | STP-DST | STP-DST + Scalable-DSC[N+R] | SOM-DST | SOM-DST + Scalable-DSC[N+R] | STAR | STAR + Scalable-DSC[N+R] |
> | --------------------- | :-----: | :-------------------------: | :-----: | :-------------------------: | :--: | :----------------------: |
> | taxi-departure        |  1.38   |            0.70             |  1.92   |            1.33             | 0.85 |           0.59           |
> | taxi-destination      |  1.20   |            0.69             |  1.80   |            1.26             | 1.01 |           0.82           |
> | taxi-leave            |  0.42   |            0.37             |  0.38   |            0.29             | 0.00 |           0.00           |
> | taxi-arrive           |  0.55   |            0.32             |  0.77   |            0.32             | 0.00 |           0.00           |
> | restaurant-name       |  2.02   |            1.46             |  3.33   |            2.83             | 1.77 |           1.46           |
> | restaurant-area       |  0.75   |            0.71             |  1.64   |            1.22             | 0.96 |           0.80           |
> | restaurant-pricerange |  1.26   |            1.03             |  1.60   |            1.37             | 1.34 |           1.01           |
> | restaurant-food       |  1.15   |            0.84             |  1.64   |            0.86             | 1.07 |           0.80           |
> | restaurant-day        |  0.24   |            0.21             |  0.35   |            0.40             | 0.33 |           0.28           |
> | restaurant-people     |  0.37   |            0.21             |  0.39   |            0.24             | 0.24 |           0.13           |
> | restaurant-time       |  0.47   |            0.40             |  1.00   |            1.01             | 0.57 |           0.58           |
> | attraction-name       |  2.46   |            1.54             |  3.52   |            2.64             | 2.21 |           1.71           |
> | attraction-area       |  1.13   |            1.13             |  1.41   |            1.37             | 1.79 |           1.79           |
> | attraction-type       |  1.83   |            1.73             |  1.68   |            1.39             | 1.60 |           1.56           |
> | hotel-name            |  1.34   |            0.94             |  3.51   |            2.23             | 1.47 |           0.93           |
> | hotel-internet        |  1.81   |            1.68             |  1.14   |            1.04             | 2.01 |           1.65           |
> | hotel-area            |  1.43   |            1.11             |  1.66   |            1.12             | 2.83 |           2.47           |
> | hotel-parking         |  1.28   |            1.12             |  1.27   |            0.86             | 2.22 |           1.99           |
> | hotel-stars           |  1.00   |            0.86             |  1.03   |            0.93             | 0.71 |           0.65           |
> | hotel-type            |  3.73   |            2.12             |  4.43   |            1.87             | 3.37 |           4.96           |
> | hotel-pricerange      |  1.68   |            1.55             |  1.79   |            1.56             | 1.58 |           1.42           |
> | hotel-day             |  0.17   |            0.01             |  0.33   |            0.33             | 0.10 |           0.06           |
> | hotel-people          |  0.37   |            0.08             |  0.42   |            0.44             | 0.20 |           0.20           |
> | hotel-stay            |  0.17   |            0.05             |  0.27   |            0.32             | 0.27 |           0.19           |
> | train-departure       |  0.63   |            0.36             |  1.11   |            0.70             | 0.66 |           0.12           |
> | train-day             |  0.32   |            0.20             |  0.86   |            0.70             | 0.25 |           0.23           |
> | train-arrive          |  0.74   |            0.66             |  1.09   |            1.01             | 0.0  |           0.00           |
> | train-destination     |  0.99   |            0.71             |  0.99   |            0.76             | 0.78 |           0.32           |
> | train-leave           |  2.04   |            1.93             |  2.23   |            2.14             | 0.0  |           0.0            |
> | train-people          |  1.39   |            0.28             |  1.07   |            0.67             | 0.73 |           0.42           |
>
> **Table1.** The error rate of each slot on MultiWOZ 2.4. (All parameter settings for this experiment are consistent with those mentioned in our paper. ）
>
> **Response4:**
>
> Thank you very much for the reviewer's suggestions. They will be valuable for our future research. We further investigated the error rates for each slot. We used the STP-DST, SOM-DST and STAR models as our primary models and used the Scalable-DSC [N+R] model to correct the predictions of the primary models. The results on the MultiWOZ 2.4 test set are shown in Table 1. It can be seen that the Scalable-DSC model effectively reduces the error rates for most slots. However, we found that slots associated with 'name' and 'type' have higher error rates in the primary model itself. Although the error rates decrease after correction, these slots still present a challenge. These will be added in the improved manuscript along with some explanations and analysis.
>
> ### **References**
>
> [1] Xie H, Su H, Song S, et al. Correctable-DST: Mitigating Historical Context Mismatch between Training and Inference for Improved Dialogue State Tracking[C]//Proceedings of the 2022 Conference on Empirical Methods in Natural Language Processing. 2022: 876-889.

---

### Official Review · Reviewer_kfrt · 2023-08-05

**Soundness:** 4

**Excitement:**

4: Strong: This paper deepens the understanding of some phenomenon or lowers the barriers to an existing research direction.

**Paper Topic And Main Contributions:**

The authors present a scalable dialogue state correction model that is agnostic to the dialogue state tracking model and yields large performance improvements for all dialogue state tracking models that it is combined with. It shows strong improvements mostly for over-predictions.

**Questions For The Authors:**

- While the performance gains in this paper and Xie et al. 2022 is impressive, it is difficult for me to understand why DSC is a necessary step. Is there any explanation for why DST with an additional dialogue state correction step performs better than without it, even when both cases receive the full dialogue history? While I understand the issue with error propagation, it's difficult to understand why there will be an error that is correctible in a future step when it was predicted wrong in the previous step. Can it be explained by adding consistency through redundancy? If so, could we just generate multiple dialogue states and take a majority vote to improve consistency? Or is there something about the correction step that is easier than the initial prediction? Related to this, lines 40-46 is incorrect as there are still many recent works that use the entire dialogue history to predict each turn's dialogue state. Also, even Xie et al. 2022 provides the entire history and also the predicted dialogue state of the previous turn, so the information provided by the dialogue history and predicted dialogue state should, in theory, be repetitive.
- Line 47: efficiency in terms of what? inference speed? If the full dialogue history is provided anyways along with the predicted dialogue state, which is the case for many of the baselines mentioned here, it cannot be speed.
- What is the usefulness of lines 93-95 of generating erroneous states and then continuing to generate a sequence of corrected states? Is this the training data formulation? This seems the same as Xie et al., 2022.
- What is meant by lines 106-108? How does STP-DST provide training data that is different from the strategies described above? I think the paragraph meant to say that the strategies are for getting incorrect predicted dialogue states, which is part of the input to the DSC model while STP-DST is a way to get data that is in the right format for the full input output of the training data for the DSC model. This is made clearer when reading the methodology and seeing Figure 2, but it's confusing when reading this part.
- I wonder if this model can actually be used to correct wrong reference dialogue states in the datasets. What are your thoughts?
- Does the proposed approach require using two models, one for the initial prediction and the correction? If so, this harms latency, ease of deployment, and adds overhead of maintaining two separate models.
- Why do you separate [N -> R] and [R->N] and not do [R+N], as in training the simulator at the same time?

**Reasons To Accept:**

- The authors build on previous work on dialogue state correction by proposing a model that is agnostic to the DST model while still yielding significant performance improvements to the models that it is paired with.
- The authors share a rigorous study with many baselines and analysis on the type of errors, clearly demonstrating the strengths and weaknesses of this approach.

**Reasons To Reject:**

- While the DSC model itself is scalable in that it is agnostic to the DST model, the steps employed for creating the templates seem to require quite a bit of manual labor. Therefore, it is questionable who scalable this approach is to new datasets in that it will require manual labor for creating templates specific to each domain.
- There are no limitations mentioned on practical aspects of using a DSC model, especially one proposed that requires two separate models: one for the initial prediction and one for the correction. This harms latency and ease of deployment, and adds additional overhead compared to a single model.
- Minor: the paper makes some incorrect claims in the introduction about the majority of DST approaches using the previous dialogue state as a compact representation of the dialogue history information when in fact most of the baselines mentioned, and even the proposed model, uses the full dialogue history *and* the previous dialogue state.

**Reproducibility:**

4: Could mostly reproduce the results, but there may be some variation because of sample variance or minor variations in their interpretation of the protocol or method.

**Reviewer Confidence:**

5: Positive that my evaluation is correct. I read the paper very carefully and I am very familiar with related work.

**Typos Grammar Style And Presentation Improvements:**

- remove Specifically in line 74
- line 84: be -> is
- line 98: during the training -> during training
- line 102: generalization ability -> generalizability
- The example in Figure 2 shows some area slots as east while some as west. I think the intended slot value was supposed to be west for all examples. It would be better to make it consistent as to reduce confusion. At least, the output of STP-DST should be east instead of west if it is to be consistent with the inference results.
- The ethics statement is empty.

---

> ### Author Rebuttal · Authors · 2023-08-29
>
> Thank you very much for your informative and constructive review. The responses to the comments are listed as follows.
>
> ### **Questions:**
>
> **Q1:**
>
> **(a) While the performance gains in this paper and Xie et al. 2022 is impressive, it is difficult for me to understand why DSC is a necessary step. Is there any explanation for why DST with an additional dialogue state correction step performs better than without it, even when both cases receive the full dialogue history? While I understand the issue with error propagation, it's difficult to understand why there will be an error that is correctible in a future step when it was predicted wrong in the previous step. Can it be explained by adding consistency through redundancy? If so, could we just generate multiple dialogue states and take a majority vote to improve consistency? Or is there something about the correction step that is easier than the initial prediction?**
>
> **(b) Related to this, lines 40-46 is incorrect as there are still many recent works that use the entire dialogue history to predict each turn's dialogue state. Also, even Xie et al. 2022 provides the entire history and also the predicted dialogue state of the previous turn, so the information provided by the dialogue history and predicted dialogue state should, in theory, be repetitive.**
>
> | Model            | Epoch | JGA(%) | Train Goal                |
> | ---------------- | :---: | :----: | :------------------------ |
> | STP-DST[N]       |  10   | 78.80  | Predicting and Correcting |
> | STP-DST          |   8   | 76.31  | Predicting                |
> | +Scalable-DSC[N] |   3   | 81.17  | Correcting                |
>
> **Table1.** Joint goal accuracy (%) on the test set of MultiWOZ2.4. [N]: The model is trained using predicted dialogue states simulated by a predictive state simulator as training data. （All parameter settings for this experiment are consistent with those mentioned in our paper. ）
>
> **Response1：**
>
> (a)Both the DST and Scalable-DSC models use complete dialogue history and dialogue state. However, their training objectives are different. DST's training objective is to track dialogue state based on the previous dialogue state and history, while Scalable-DSC's training objective is to rectify errors in the dialogue state based on the dialogue history, which empowers the Scalable-DSC model with correction capability. Furthermore, the Correctable Dialogue State Tracking (Correctable-DST) model's training objective is to predict the dialogue state of the current turn while correcting errors in the dialogue state from the previous turn. We attempted to utilize the approach of Correctable-DST, enabling the STP-DST model to simultaneously predict and correct dialogue states. The results are shown in Table 1. Under the same parameter settings, we found that Scalable-DSC achieves the best performance with just 3 epochs of training, while STP-DST requires only 8 epochs of training. Therefore, correcting errors is easier than making initial predictions, and making predictions alone is easier than simultaneously predicting and correcting dialogue states. The experimental results also indicate that the performance of DST with an additional dialogue state correction step is better than without it. These will be added in the improved manuscript along with some explanations and analysis.
>
> When errors occur in the previous dialogue state, it indicates a discrepancy between the information in the dialogue history and the information in the dialogue state. Therefore, the dialogue history can be used as evidence for detecting and correcting errors in the dialogue state. The approach of generating multiple dialogue states and conducting majority voting to enhance consistency is feasible, but this will significantly impact decoding speed[3].
>
> (b) During the training phase, these DST models take the actual previous dialogue state as input. At this stage, the information provided by the dialogue history and the predicted dialogue state is theoretically redundant. However, during the inference phase, the predicted previous dialogue state is used, which may introduce errors compared to the dialogue history. This is because we cannot guarantee the complete accuracy of the predicted dialogue state. Therefore, in the inference phase, the information from the dialogue history and the predicted dialogue state may not be entirely consistent.
>
> In lines 40-46, the wording compromises readability. We appreciate your valuable feedback and will address this in the subsequent version of the paper. The revised version is as follows: "Some DST methods choose the previous turn's dialogue state as part of the input to track the user's goal".
>
> **Q2: Line 47: efficiency in terms of what? inference speed? If the full dialogue history is provided anyways along with the predicted dialogue state, which is the case for many of the baselines mentioned here, it cannot be speed.**
>
> **Response2:**
>
> We are grateful to the reviewers for pointing out the errors in our paper. Our intended message was that using the dialogue state from the previous turn, in conjunction with the dialogue utterance from the current turn, leads to more efficient prediction of the current turn's dialogue state in the DST model, resulting in better performance[6]. We will carefully adjust this statement based on the context of the paper during the revision process.
>
> **Q3: What is the usefulness of lines 93-95 of generating erroneous states and then continuing to generate a sequence of corrected states? Is this the training data formulation? This seems the same as Xie et al., 2022.**
>
> | Model              | MultiWOZ 2.4 | Generation Type                                            |
> | ------------------ | :----------: | ---------------------------------------------------------- |
> | STP-DST            |    76.31     | -                                                          |
> | +Scalable-DSC[N+R] |    81.62     | generate erroneous states+generate corrected states (ours) |
> | +Scalable-DSC[N+R] |    80.90     | generate corrected states                                  |
>
> **Table 2.** Joint goal accuracy (%) on the test set of MultiWOZ2.4. "[N+R]": First using [N] strategy and then the [R] strategy to train the Scalable-DSC model in stages. （All parameter settings for this experiment are consistent with those mentioned in our paper. ）
>
> **Response3:**
>
> We believe that when correcting dialogue state, identifying the wrong slot first before correction is more beneficial in reducing the inference burden of the model than direct correction. This idea has been validated as shown in Table 2. On the MultiWOZ 2.4 dataset, when correcting the predictions of the STP-DST model, using Scalable-DSC to only generate corrected dialogue state sequences achieved a 80.90% Joint Goal Accuracy (JGA), while using the method proposed in this paper achieved an 81.62% JGA. The experimental results show that our proposed method achieved a 0.72% improvement, proving that identifying the wrong slot first before correction is more beneficial in reducing the inference burden of the model than direct correction. Compared to Correctable-DST[2], our proposed method infers both the wrong slot and value, with a finer granularity. These will be added in the improved manuscript along with some explanation and analysis.
>
> **Q4: What is meant by lines 106-108? How does STP-DST provide training data that is different from the strategies described above? I think the paragraph meant to say that the strategies are for getting incorrect predicted dialogue states, which is part of the input to the DSC model while STP-DST is a way to get data that is in the right format for the full input output of the training data for the DSC model. This is made clearer when reading the methodology and seeing Figure 2, but it's confusing when reading this part.**
>
> **Response4:** The predicted state data provided by STP-DST in lines 106-108 serves as the training data required for Strategy 2. Your understanding of the meaning is accurate.
>
> **Q5: I wonder if this model can actually be used to correct wrong reference dialogue states in the datasets. What are your thoughts?**
>
> | Model    | MultiWOZ 2.4(%) |
> | -------- | :-------------: |
> | SOM-DST  |      66.78      |
> | SOM-DST* |      69.68      |
>
> **Table 3.** Joint goal accuracy (%) on the test set of MultiWOZ2.4. * refer to training the model by incorporating the pseudo-labels generated after correcting the training set using the Scalable-DSC  model along with the original training set. （All parameter settings for this experiment are consistent with those mentioned in our paper. ）
>
> **Response 5:**
>
> Thank you very much for the reviewer's suggestions. They will be valuable for our future research. We have attempted to use Scalable-DSC to correct for erroneous dialogue states within the dataset, taking inspiration from the experimental setup of ASSIST[4]. They considered the validation set to be the cleanest approximation of noise-free data, and likewise we used the validation set to train the Scalable-DSC model. This model was then used to correct mislabelled data within the training set. The pseudo-labelled data generated by Scalable-DSC was then used to train the SOM-DST model. The experimental results are presented in Table 3. On the MultiWOZ 2.4 dataset, the Joint Goal Accuracy (JGA) improved to 69.68% compared to the baseline result of 66.78% achieved by training the SOM-DST model solely on the original training data, an improvement of 2.9%.  Although this performance falls short of the ASSIST performance, it demonstrates the preliminary potential of the Scalable-DSC model to correct erroneous mislabelled data within the dataset. All parameter settings of the current experiment are consistent with those of the paper. We find this a very intriguing concept and are actively engaged in further exploration.
>
> **Q6: Does the proposed approach require using two models, one for the initial prediction and the correction? If so, this harms latency, ease of deployment, and adds overhead of maintaining two separate models.**
>
> | model              | MultiWOZ 2.0  | MultiWOZ 2.1  | MultiWOZ 2.2  | MultiWOZ 2.3  | MultiWOZ 2.4 |
> | ------------------ | :-----------: | :-----------: | :-----------: | :-----------: | :----------: |
> | SOM-DST            |     51.38     |     52.57     |     53.27     |     55.50     |    66.78     |
> | +Scalable-DSC[N+R] | 68.61(+17.23) | 66.83(+14.26) | 67.14(+13.87) | 67.23(+11.73) | 74.37(+7.59) |
> | STAR               |     54.53     |     56.36     |     60.49     |     65.87     |    73.62     |
> | +Scalable-DSC[N+R] | 68.51(+13.98) | 69.02(+12.66) | 71.33(+10.84) | 72.50(+6.63)  | 78.91(+5.29) |
> | STP-DST            |     55.51     |     55.85     |     57.08     |     63.79     |    76.31     |
> | +Scalable-DSC[N+R] | 66.10(+10.59) | 69.13(+13.28) | 71.83(+14.75) | 74.46(+10.67) | 81.62(+5.31) |
>
> **Table4.**  Joint goal accuracy (%) on the test set of MultiWOZ2.4. "[N+R]": First using [N] strategy and then the [R] strategy to train the Scalable-DSC model in stages. (All parameter settings for this experiment are consistent with those mentioned in our paper. ）
>
> **Response6:**
>
> We found it necessary to use two models: one for initial prediction and one for correction. This adds overhead, but the performance gains are significant, e.g. on the MultiWOZ 2.0 dataset Scalable-DSC[N+R]  improved the joint target accuracy of SOM-DST by 17.23%, as shown in Table 4.
>
> **Q7: Why do you separate [N -> R] and [R->N] and not do [R+N], as in training the simulator at the same time?**
>
> | Model               | MultiWOZ 2.4(%) |
> | ------------------- | :-------------: |
> | STP-DST             |      76.31      |
> | +Scalable-DSC[N->R] |      81.62      |
> | +Scalable-DSC[R->N] |      80.52      |
> | +Scalable-DSC[R+N]  |      80.10      |
>
> **Table5.** Joint goal accuracy (%) on the test set of MultiWOZ2.4. "[N->R]": First using [N] strategy and then the [R] strategy to train the Scalable-DSC model in stages. "[R->N]" : First using the [R] strategy and then the [N] strategy to train the Scalable-DSC model in stages. "[R+N]" : Mix the data generated by N strategy with that generated by R strategy for model training. (All parameter settings for this experiment are consistent with those mentioned in our paper. ）
>
> **Response7:**
>
> Training the Scalable-DSC model separately using the predicted dialogue state from other DST models as training data can lead to a better adaptation of the Scalable-DSC model to the distribution of error types in the validation and test sets predicted by other DST models. The use of R+N leads to an inconsistent data distribution of error types between the training and test sets of the Scalable-DSC model. Therefore, we adopt a staged training approach. We conducted experiments to validate this idea, and the experimental results are shown in Table 5. It can be observed that when the data from the noise simulator is combined with the data predicted by the DST model to train the Scalable-DSC model, its correction performance is inferior to the phased training Scalable-DSC model. These will be added in the improved manuscript along with some explanation and analysis. And if the paper is accepted, we will make the code publicly available on GitHub for verification purposes.
>
> ### **Reasons To Reject:**
>
> **Q1: While the DSC model itself is scalable in that it is agnostic to the DST model, the steps employed for creating the templates seem to require quite a bit of manual labor. Therefore, it is questionable who scalable this approach is to new datasets in that it will require manual labor for creating templates specific to each domain.**
>
> **Response1:**
>
> The structural templates we constructed were generated using a heuristic converter that supports interconversion between dialogue states and natural language sequences, and the heuristic converter we used was extracted and refactored from the work of DS2 [5]. The Appendix A4 section in DS2 describes how to extend the template generation approach to new domain data, It explains that the developer only needs to rewrite the dialogue heuristic converter between states and summaries is sufficient, similarly extending our proposed method to new domain data requires only a slight modification of the code for the heuristic converter, which can be implemented by a Python developer in just a few hours, ensuring that our correction method is easily scalable to other datasets. If our paper is accepted, we will publish the code for the heuristic converter on github.
>
> **Q2：There are no limitations mentioned on practical aspects of using a DSC model, especially one proposed that requires two separate models: one for the initial prediction and one for the correction. This harms latency and ease of deployment, and adds additional overhead compared to a single model.**
>
> **Response2:**
>
> Thank you very much for your suggestions. This helps prevent confusion for others reading the paper. We have addressed the relevant issues in Response 6 of the aforementioned Section Questions. We found it necessary to use two models: one for initial prediction and one for correction. This adds overhead, but the performance gains are significant, e.g. on the MultiWOZ 2.0 dataset Scalable-DSC[N+R]  improved the joint target accuracy of SOM-DST by 17.23%, as shown in Table 4.
>
> **Q3：Minor: the paper makes some incorrect claims in the introduction about the majority of DST approaches using the previous dialogue state as a compact representation of the dialogue history information when in fact most of the baselines mentioned, and even the proposed model, uses the full dialogue history \*and\* the previous dialogue state.**
>
> **Response3:**
>
> Thank you very much for your suggestions. They will help us improve the quality of the manuscript. We have addressed the relevant issues in Response 1 of the aforementioned Section Questions. In lines 40-46, the wording compromises readability.  We will address this in the subsequent version of the paper. The revised version is as follows: "Some DST methods choose the previous turn's dialogue state as part of the input to track the user's goal".
>
> **We appreciate the comments from the reviewers that have contributed to improving the readability of the paper. We have considered all suggestions and corrected grammar errors. We will optimize the example of Figure 2 in the improved manuscript.**
>
> ## **References**
>
> [1] Tian X, Huang L, Lin Y, et al. Amendable Generation for Dialogue State Tracking[C]//Proceedings of the 3rd Workshop on Natural Language Processing for Conversational AI. 2021: 80-92.
>
> [2] Xie H, Su H, Song S, et al. Correctable-DST: Mitigating Historical Context Mismatch between Training and Inference for Improved Dialogue State Tracking[C]//Proceedings of the 2022 Conference on Empirical Methods in Natural Language Processing. 2022: 876-889.
>
> [3]  Won S, Kwak H, Shin J, et al. BREAK: Breaking the Dialogue State Tracking Barrier with Beam Search and Re-ranking[C]//Proceedings of the 61st Annual Meeting of the Association for Computational Linguistics (Volume 1: Long Papers). 2023: 2832-2846.
>
> [4] Ye F, Feng Y, Yilmaz E. ASSIST: Towards Label Noise-Robust Dialogue State Tracking[C]//Findings of the Association for Computational Linguistics: ACL 2022. 2022: 2719-2731.
>
> [5] Shin J, Yu H, Moon H, et al. Dialogue Summaries as Dialogue States (DS2), Template-Guided Summarization for Few-shot Dialogue State Tracking[C]//Findings of the Association for Computational Linguistics: ACL 2022. 2022: 3824-3846.
>
> [6] Liu, Hong et al. “Revisiting Markovian Generative Architectures for Efficient Task-Oriented Dialog Systems.” *ArXiv* abs/2204.06452 (2022): n. pag.

---

### Meta-Review · Area_Chair_3oE3 · 2023-09-19

**Recommendation:** 4

**Metareview:**

The paper presents a scalable dialogue state error correction approach, and when paired with DST models (to which the DSC is agnostic), the system achieves state-of-the-art performance on the MultiWOZ datasets. The empirical studies presented in the paper is considered as rigorous, with room for improvement by expanding experiments towards non-MultiWOZ DST datasets. Also the practical limitations of employing a DSC component should be discussed more thoroughly in the paper.

---

### Decision · Program_Chairs · 2023-10-07

**Decision:**

Accept-Main

**Comment:**

The paper presents a scalable dialogue state error correction approach, and when paired with DST models (to which the DSC is agnostic), the system achieves state-of-the-art performance on the MultiWOZ datasets. The empirical studies presented in the paper is considered as rigorous, with room for improvement by expanding experiments towards non-MultiWOZ DST datasets. Also the practical limitations of employing a DSC component should be discussed more thoroughly in the paper.